# Precision Nutrition to Activate Thermogenesis as a Complementary Approach to Target Obesity and Associated-Metabolic-Disorders

**DOI:** 10.3390/cancers13040866

**Published:** 2021-02-18

**Authors:** Marina Reguero, Marta Gómez de Cedrón, Sonia Wagner, Guillermo Reglero, José Carlos Quintela, Ana Ramírez de Molina

**Affiliations:** 1Molecular Oncology Group, Precision Nutrition and Health, IMDEA Food Institute, CEI UAM + CSIC, Ctra. de Cantoblanco 8, 28049 Madrid, Spain; marina.reguero@imdea.org (M.R.); sonia.wagner@imdea.org (S.W.); 2NATAC BIOTECH, Electronica 7, Alcorcón, 28923 Madrid, Spain; jcquintela@natacgroup.com; 3Medicinal Gardens SL, Marqués de Urquijo 47, 28008 Madrid, Spain; 4Production and Characterization of Novel Foods Department, Institute of Food Science Research CIAL, CEI UAM + CSIC, 28049 Madrid, Spain; guillermo.reglero@imdea.org

**Keywords:** precision nutrition, bioactive compounds, metabolic diseases, thermogenesis

## Abstract

**Simple Summary:**

Regarding the pandemic of obesity and chronic diseases associated to metabolic alterations that occur nowadays worldwide, here, we review the most recent studies related to bioactive compounds and diet derived ingredients with potential effects to augment the systemic energy expenditure. We specifically focus in two processes: the activation of thermogenesis in adipose tissue and the enhancement of the mitochondrial oxidative phosphorylation capacity in muscles. This may provide relevant information to develop diets and supplements to conduct nutritional intervention studies with the objective to ameliorate the metabolic and chronic inflammation in the course of obesity and related disorders.

**Abstract:**

Obesity is associated to increased incidence and poorer prognosis in multiple cancers, contributing to up to 20% of cancer related deaths. These associations are mainly driven by metabolic and inflammatory changes in the adipose tissue during obesity, which disrupt the physiologic metabolic homeostasis. The association between obesity and hypercholesterolemia, hypertension, cardiovascular disease (CVD) and type 2 diabetes mellitus (T2DM) is well known. Importantly, the retrospective analysis of more than 1000 epidemiological studies have also shown the positive correlation between the excess of fatness with the risk of cancer. In addition, more important than weight, it is the dysfunctional adipose tissue the main driver of insulin resistance, metabolic syndrome and all cause of mortality and cancer deaths, which also explains why normal weight individuals may behave as “metabolically unhealthy obese” individuals. Adipocytes also have direct effects on tumor cells through paracrine signaling. Downregulation of adiponectin and upregulation of leptin in serum correlate with markers of chronic inflammation, and crown like structures (CLS) associated to the adipose tissue disfunction. Nevertheless, obesity is a preventable risk factor in cancer. Lifestyle interventions might contribute to reduce the adverse effects of obesity. Thus, Mediterranean diet interventional studies have been shown to reduce to circulation inflammatory factors, insulin sensitivity and cardiovascular function, with durable responses of up to 2 years in obese patients. Mediterranean diet supplemented with extra-virgin olive oil reduced the incidence of breast cancer compared with a control diet. Physical activity is another important lifestyle factor which may also contribute to reduced systemic biomarkers of metabolic syndrome associated to obesity. In this scenario, precision nutrition may provide complementary approaches to target the metabolic inflammation associated to “unhealthy obesity”. Herein, we first describe the different types of adipose tissue -thermogenic active brown adipose tissue (BAT) versus the energy storing white adipose tissue (WAT). We then move on precision nutrition based strategies, by mean of natural extracts derived from plants and/or diet derived ingredients, which may be useful to normalize the metabolic inflammation associated to “unhealthy obesity”. More specifically, we focus on two axis: (1) the activation of thermogenesis in BAT and browning of WAT; (2) and the potential of augmenting the oxidative capacity of muscles to dissipate energy. These strategies may be particularly relevant as complementary approaches to alleviate obesity associated effects on chronic inflammation, immunosuppression, angiogenesis and chemotherapy resistance in cancer. Finally, we summarize main studies where plant derived extracts, mainly, polyphenols and flavonoids, have been applied to increase the energy expenditure.

## 1. Introduction

According to the World-Health-Organization obesity and chronic metabolic diseases are augmenting alarmingly. The increase in high-fat-saturated diets and the sedentary lifestyle of western countries have become the main cause of obesity and type 2 diabetes which are becoming pandemic with many other metabolic disorders associated (Figure 1). During obesity, the adipose tissue (AT) becomes dysfunctional, promoting a pro-inflammatory and insulin-resistant environment that contributes to metabolic alterations of key metabolic organs, such as liver, pancreas, or muscle, implicated in the systemic energetic homeostasis. This imbalance contributes to the development of type 2 diabetes mellitus (T2DM) and a systemic proinflammatory state, which in turn may lead to metabolic alterations with multi-organ damage [1,2,3].

The most straightforward way to prevent obesity is to limit calorie intake or to increase the burning of calories through exercise and physical activity. However, majority of subjects fail in their attempts to accomplish these aims. Thus, at present, the only method to combat excessive obesity is through surgery, as many drugs have failed causing serious and deleterious side effects. Nonetheless, it is an invasive method not available for all patients which also requires long time of recovery. Therefore, it has gained importance to investigate in complementary approaches in the frame of Precision Nutrition to alleviate obesity and associated metabolic alterations (Figure 1). As AT contributes to the control of the systemic energetic balance, strategies to increase the thermogenic potential of AT to augment the energetic expenditure (EE) may be relevant in the control of metabolic alterations. In this regard, the crosstalk between adipose tissue with other metabolic organs such as skeletal muscle is another complementary strategy to be explored [4,5,6,7,8,9,10,11,12,13,14,15].

As indicated, the current obesity pandemic results from a physiological imbalance in which energy intake chronically exceeds EE. Approaches designed to increase EE have been demonstrated in animal models mainly oriented to stimulate the adaptive thermogenesis. At the cellular level, thermogenesis is mainly achieved by increasing the uncoupling of the oxidative phosphorylation from adenosine triphosphate (ATP) generation by the uncoupling protein 1 (UCP1), a tissue-specific protein highly expressed in mitochondria of brown adipose tissue (BAT). Physiological activation of BAT is mainly mediated through β-adrenergic receptors (AR) activation. This way, considerable efforts over the past 5 decades have been oriented to identify AR agonists capable of safely achieve a net negative energy balance while avoiding unwanted cardiovascular side effects. Recent discoveries of other BAT futile cycles based on creatine and succinate have provided additional targets. Cold and exercise are also physiological key players with potential to increase the EE, and developing strategies are being investigated with the aim to mimic these processes by mean of the use of bioactive compounds.

Future studies will address technical challenges such as how to accurately measure individual tissue thermogenesis in humans; how to safely activate BAT and thermogenesis in other metabolic organs, mainly white adipose tissue (WAT) and skeletal muscle; and how to sustain a negative energy balance over many years of treatment.

## 2. Methodology of Searching

The methodology used for the selection of the references included in this review was based on the searching algorithm presented below. For this, two different investigators searched in PubMed and Web of Science databases from 13 of April to 30 of October 2020, for studies that evaluated the principal targets that could be modulated by diet/bioactive compounds regarding thermogenesis in muscle and adipose tissue, and also, for studies that analyzed the effect of the five plants represented in this review regarding the modulation of thermogenesis in different studies.

Thus, we used our systematic criteria of searching for articles and reviews no older than 10 years that include in their title or in their abstract these following concepts:

[Mitochondrial thermogenesis OR white adipose tissue browning process OR brown adipocyte thermogenesis activation OR beige adipocyte thermogenesis OR Brown adipogenesis OR thermogenic obesity treatment OR cold-induced thermogenesis OR leptin thermogenesis OR energy expenditure and thermogenesis OR muscle non-shivering thermogenesis OR exercise contribution in energy expenditure and thermogenesis OR AMPK thermogenesis activation OR SIRT1 thermogenesis activation PGC1a thermogenesis activation AND/OR precision nutrition OR nutrigenomics OR nutrigenetics OR Resveratrol OR Pomegranate OR Silymarin OR Soy OR Ginkgo OR natural extracts.]

With those keywords, 1724 articles were found (once duplicates were excluded) on both websites. From this, we excluded 563 articles with no full-text assessed for eligibility. The abstract of the remaining 1171 articles where read, and finally 435 papers and reviews were included. 214 were included based on their relevant information related to thermogenesis, browning and energy expenditure.

We also searched the reference lists of these studies, to identify potentially eligible studies that were not included in the initial search. Disagreements in the searching procedure between the two independent investigators were resolved through scientific consensus.

## 3. Addressing Obesity through Precision Nutrition: Nutrigenetics and Nutrigenomics

It is noteworthy that the relationship between diet and health is a matter of great interest, and nutrition and diet derived ingredients can also be considered as tools to prevent and/or to treat different chronic diseases. It has been proved that general macronutrient recommendations do not affect individuals in the same way, as these recommendations do not take into consideration the genetic heterogenicity between individuals, such as genetic variants (single nucleotide polymorphisms or SNPs), nor the nutritional and metabolic status of individuals (such as dyslipidemia, insulin resistance, metabolic syndrome, cardiovascular disease, and cancer) that shape the final metabolic effects of diet derived ingredients. Therefore, the knowledge of interactions between genome and nutrients at the molecular level has led to the development of ”Nutritional Genomics”, which involves the sciences of nutrigenomics and nutrigenetics [16,17].

In a deeper meaning, nutrigenetics focuses on inherited or acquired mutations within genes and the interaction between bioactive compounds and ingredients from diet with the genetic susceptibilities to metabolic disorders. On the other hand, nutrigenomics includes several disciplines for the study of the dietary effects on genome stability regarding deoxyribonucleic acid (DNA) damage, epigenomic alterations, ribonucleic acid (RNA) expression, and protein expression, which will affect the individuals’ metabolic health. Thus, the key goal of nutritional genomics is the use of this information meaningfully to provide trustable personalized dietary recommendations for specific health and disorder prevention outcomes [16,18].

Precision nutrition is considered as a two-way route integrating by one hand the knowledge of the diversity in the genomes which also influences nutrient bioavailability and metabolism, and by the other hand, the knowledge of how nutrients may affect the expression of genes in critical metabolic pathways [19]. In addition, Precision nutrition considers additional factors related to lifestyle such as diet, exercise, alcohol consumption, as well as the nutritional and metabolic status of the individual. Based on this, the final objective of precision nutrition is to develop personalized nutrition interventions for health and disease prevention or treatment.

The development of multi-omics sciences and bioinformatics tools, together with the sequencing of the human genome, has contributed to the development of effective personalized nutritional interventions. Only the scientific knowledge of individuals´ genomic information combined with the molecular characterization of diet ingredients and the elucidation of their molecular mechanisms of action may provide effective personalized nutritional interventions [7,20,21].

Several studies have revealed genetic associations to the predisposition to obesity and/or type 2 diabetes and associated comorbidities [22]. For instance Alpha-Ketoglutarate Dependent Dioxygenase (*FTO*), Insulin Like Growth Factor 2 MRNA Binding Protein 2 *(IGF2BP2),* Brain Derived Neurotrophic Factor *(BDNF),* Melanocortin 4 Receptor *(MC4R)*, and Apolipoprotein *(APO-A)* gene family, among others, have been shown to present SNPs associated with those disorders [23,24,25,26,27,28,29] The genetic susceptibility to obesity is also influenced by an obesogenic environment for its phenotypic expression (nutrigenetics) (Figure 1).

In this scenario, the identification of brown and beige adipocytes in adult humans has opened the possibility of using thermogenesis and browning of WAT as a strategy to prevent and treat obesity and related metabolic alterations diseases, which can compensate genetic polymorphisms and/or mutations in a personalized manner [30,31]. This way, it is possible to develop nutritional interventions by mean of the use of bioactive compounds and/or ingredients from diet to decrease fat storage -by mean of the induction of mitochondrial biogenesis, by increasing lipolysis coupled to fatty acid oxidation-, and to promote the brown adipogenesis in WAT to compensate the genetic and environmental factors implicated in the development of metabolic alterations.

## 4. Adipose Tissue Features and Fate: The Browning Process

In the last years it has been accepted that AT is not a passive organ, instead, it is a dynamic endocrine organ implicated in the control of the energy homeostasis and inflammation among other processes [32]. Therefore, AT as an endocrine organ releases a variety of adipokines (adipose-associated cytokines) which affect the functioning of different organs and tissues including the liver, pancreas, muscle, brain and immune system. 

There are two types of AT differing in their mesenchymal progenitor origin, their functionality and their distribution within the body (Figure 2). The WAT constitutes the body’s main energy reserve, storing and mobilizing triglycerides depending on the requirement of the different peripheral tissues. WAT can be considered as an organ formed by a cellular component (preadipocytes, mature adipocytes, fibroblasts, immune infiltrated cells, and endothelial cells), and a non-cellular component (mainly stroma and matrix fibers). Mature white adipocytes synthesize and secrete adipokines to regulate the energy balance and the systemic glucose and lipid metabolisms [33,34]. This way, WAT senses the body’s energetic status and responds by storing fats in the form of triacylglycerides, or by releasing glycerol and fatty acids to be used as energy sources when required.

On the other hand, BAT is a thermogenic organ responsible for maintaining the thermal homeostasis through dissipating large amounts of chemical energy as heat. The existence of BAT in infants has been known for a long time, but the demonstration of metabolically active BAT in adults was only documented recently [35] The main characteristic of BAT is the high levels of mitochondria which provides high capacity to oxidize fatty acids and glucose [36]. This way, BAT may contribute to dissipate the excess of reductive power into heat in conditions of excess of glucose and fatty acids, preventing the appearance of insulin resistance and/or metabolic unhealthy obesity.

BAT has been shown to display a specific expression profile promoting lipolysis and mitochondrial uncoupling for energy dissipation (Figure 3) which contributes to alleviate the excessive metabolic stress in chronic diseases such as obesity, insulin resistance, cardiovascular disease (CVD) and cancer [37,38].

### 4.1. White and Brown Adipose Tissue Biogenesis

Adipogenesis is a process that starts during development and continues throughout life. Differentiating progenitors accumulate small lipid droplets in a process guided with the developing capillary networks. Eventually these progenitors mature to form terminal differentiated adipocytes [39]. 

Nevertheless, WAT and BAT are formed by different mesenchymal progenitors, with BAT cells coming from progenitors common to the skeletal muscle tissue which are positive for myogenic factor 5 (Myf5), whereas WAT cells come from progenitors lacking this factor. Hence, as we show in Figure 2, two different types of mesenchymal precursors give rise to two different types of AT that can be find in the body. Importantly, adipogenesis is controlled by distinct transcription factors and proteins to determine the fate of preadipocytes to WAT or BAT. For instance, PR domain containing 16 (*PRDM16*), Peroxisome Proliferator Activated Receptor Gamma Coactivator 1 Alpha (PGC1a), Carnitine Palmitoyltransferase 1A (*CPT1a*) and *UCP1* are key factors promoting BAT adipogenesis and differentiation with increased mitochondrial biogenesis and fatty acid oxidation capacity. On the other hand, Peroxisome Proliferator Activated Receptor Gamma (*PPARG*), CCAAT Enhancer Binding Protein Alpha (*C/EBPα*), Sterol Regulatory Element Binding Transcription Factor 1 (*SREBP1*) and Fatty Acid Binding Protein 4 (*FABP4)* are implicated in WAT differentiation and lipid storage. Importantly, adipogenesis have been demonstrated to be modulated by external inductors such as cold exposure or nutritional agents [37,40,41].

Recently, it has been discovered that under specific circumstances, such as cold exposure, physical exercise or fasting, white adipocytes can reprogram their metabolism to behave as brown adipocytes in a process called browning. Similar to brown adipocytes, these “beige” adipocytes are able to dissipate energy in the form of heat, and therefore, may contribute to the energetic homeostasis together with BAT. Activating the browning process could be an alternative therapeutic strategy to increase EE counteracting the development of obesity and associated metabolic diseases [42,43].

### 4.2. Adipose Tissue Browning

Beige adipocytes, despite sharing some features of both brown and white adipocytes, are a unique type of adipocytes characterized by a distinct gene expression profile and adult origin compared to white or brown adipocytes [44,45]. They have multilocular lipid droplets, higher mitochondrial content with higher levels of UCP1 compared to white adipocytes, sharing a functional and phenotypic thermogenic capacity similar to that of brown adipocytes (Figure 3) [46]. Beige and brown adipocytes use uncoupling proteins (UCPs) from the inner membrane to dissipate energy as heat. All in all, brown and beige adipocytes contribute to the energetic homeostasis balancing storage and burning of fats and glucose. UCP1 is responsible for cold- and diet-induced thermogenesis and it is directly implicated in the phenotypic change of white adipocytes into beige adipocytes [43,44].

Together with UCPs, there are many different molecules implicated in the differentiation process to BAT and/or in the browning process of WAT (Figure 3). For instance, adenosine monophosphate activated protein kinase (AMPK) activation inhibits acetyl-CoA carboxylase diminishing fatty acid synthesis, and it also promotes the uptake of extracellular glucose and fatty acids for further oxidation at mitochondria [47,48]. Moreover, it also enhances mitochondrial function by directly phosphorylating and activating PGC1α [49], a master transcriptional regulator of genes involved in mitochondrial biogenesis and oxidative metabolism [50]. In addition, AMPK, through the activation of Sirtuin 1 (*Sirt1*), another important mitochondrial activator (Figure 3) [51], promotes the differentiation process into brown and beige, increasing the expression levels of UCP1.

The PPAR family includes a group of nuclear transcription factors regulating cellular differentiation, development, energy metabolism, and tumorigenesis [52]. There are three types of PPARs: PPARα, δ and γ. PPARγ is mainly expressed in WAT, internal organs and macrophages. In mature white adipocytes, it regulates the expression of genes involved in free fatty acid uptake and triglyceride synthesis. PPARγ also promotes browning due to the AMPK/Sirt1-dependent PPARγ deacetylation, which is required to recruit the BAT programmer coactivator PRDM16, leading to the selective induction of BAT genes like cell death-inducing DNA fragmentation factor-α-like effector A (CIDEA), while inhibiting typical WAT markers (Figure 3) [53,54]. PPARα also controls the transcription of this essential gene, which interacts with PGC1α to provide the machinery necessary for the differentiation of the beige adipocyte [7,55]. Reduction of AMPK signaling has been demonstrated to reduce PRDM16 activation, diminishing brown adipogenesis and promoting fibrogenesis (Figure 2) [53].

Another key factor implicated in the browning process is fibroblast Growth Factor 21 (FGF21), which is produced by BAT and beige cells, and is released into plasma where it acts as an autocrine/paracrine factor increasing the expression of UCP1 and other genes such as CIDEA to promote thermogenesis and EE [56,57]. Moreover, Irisin, a hormone produced by skeletal muscle in response to exercise, is also a potent inducer of WAT browning, highlighting the important crosstalk between these two metabolic organs (Figure 4) [58]. Therefore, targeting the expression of those genes through phytochemicals may provide a therapeutic potential for the treatment of obesity and its related disorders.

## 5. Thermogenesis within Adipose Tissue, Inductors and Their Implications in Obesity Disorders

As explained before, BAT and beige cells have the capacity to burn off nutrients upon specific external conditions and therefore to control the energy balance in the body. When mitochondrial electron chain is uncoupled from the ATP final production, the reductive power may be dissipated into heat, a process called thermogenesis [59]. Thus, thermogenesis is the ability to generate heat in the body due to metabolic reactions which, upon correct thermal homeostasis, allows mammals (like humans) to control their body temperature as a static value of approximately 36–37 °C.

Regarding this concept, there are two different types of thermogenesis, on the one hand, the involuntary activation of skeletal muscle movement which produces heat, a process called shivering; and, on the other hand, the activation of cell metabolism in BAT and skeletal muscle dissipating heat upon mitochondrial ATP uncoupling or upon a calcium flux through the sarcoplasmic reticulum in skeletal muscle fibers, a process called non-shivering thermogenesis [60]. In this context, thermogenesis can be induced by cold exposure and its consequent adrenergic stimulation [61], but also it can be induced by diet and/or by mean of thermogenic dietary supplements [7,8]. Thus, dissipating energy by enhancement thermogenesis via increasing BAT activity or mass within beige cells in WAT is one of the new approaches for the prevention and/or treatment of obesity related metabolic disorders [9,10].

It should be noted that activation of thermogenesis not only induces weight loss, but also has a positive impact on other metabolic alterations such as insulin and leptin resistance, hepatic steatosis, hyperlipidemia, hyperglycemia, hypercholesterolemia and hypertriglyceridemia, likely due to the ability of thermogenic adipocytes to uptake lipids and glucose from the circulation. Thus, it has been proposed, mostly based on in vitro and in vivo studies, that BAT may help to prevent or reverse excessive adiposity and to improve glucose and triglyceride clearance from blood. However, direct and convincing evidence from human studies is still lacked due to its relatively recent discovery [10,11].

### 5.1. Molecules and Processes Implicated in Mitochondrial Thermogenesis

In BAT and beige thermogenesis, the main operators are mitochondria. BAT has more and larger mitochondria than WAT, as well as higher expression of fatty acid oxidation-related enzymes. This fact gives BAT the appearance of being brownish in microscopic images because of the iron-containing heme cofactor in the mitochondrial enzyme cytochrome oxidase [62]. Thus, this suggests that BAT is a more prepared tissue for dissipating and wasting energy than for storing it. One of the key proteins here is UCP1 (Figure 3), which is located in the inner membrane and causes a proton leak across it, dissipating the electrochemical gradient into heat [59,63,64]. However, not everything regarding thermogenesis is UCP1 dependent, there are many other signaling pathways implicated in the activation of thermogenesis. Thus, the metabolic axis Sirt1/PGC1α/AMPK pathway has been described to promote the correct functioning of mitochondrial thermogenesis, implicating downstream effectors including mitochondrial transcription Factor A (*TFAM*) factor and CPT1α [37,50].

The in vivo treatment of mice born from obese parents with the AMPK activators metformin or 5-aminoimidazole-4-carboxamide ribonucleotide (AICAR) increases brown adipocyte progenitor cells and BAT weight, as well as WAT thermogenesis [53]. In addition, experimental reduction of AMPK in WAT, leads to reduced levels of PGC1α and mitochondrial markers, causing lipid and glucose accumulation and the promotion of liver steatosis and insulin resistance [48]. Hence, activating AMPK in adipocytes can increase browning and non-shivering thermogenesis, promoting the uptake and oxidation of glucose, fatty acids, and triglycerides, and potentially reducing the risks associated with obesity.

AMPK activation is affected by levels of reactive oxygen species (ROS) from mitochondria. Mitochondrial ROS (mtROS) influences the function of key thermogenic proteins in adipocytes. Succinate is the genuine source of thermogenic ROS in brown and beige adipocytes, being superoxide anion production proportional to UCP1-related uncoupled respiration. Upon a balanced diet, the limited ATP synthesis in BAT, due to the uncoupled respiration, maintains an elevated proton gradient and electron flux, which controls excessive mtROS accumulation. On the contrary, when there is an excess on the intake of nutrients, such as normally happens in obesity disorders, there is an enhanced mitochondrial redox pressure which may lead to excessive mtROS accumulation. Thus, when the functional redox threshold in mitochondria is overcome, mtROS disrupt the interaction between insulin receptor (IR) and insulin receptor substrate (IRS) causing insulin resistance and the impairment of adipogenesis, thermogenesis and adipocyte function [65,66,67]. Mitochondrial superoxide dismutase levels have been shown to protect against impaired thermogenesis, by reducing the levels of superoxide [65].

Thermogenesis is also depending on the mitochondrial fusion and fission balance. The remodeling of the mitochondrial structure is dynamic and sensitive to metabolic signals. The fusion process is regulated by nucleotide guanosine triphosphate hydrolase (GTPase) proteins such as mitofusin 1 and 2, and mitochondrial dynamin like GTPase (OPA1). Mitochondrial fusion allows the repolarization of membranes diminishing ROS, which is necessary for mitochondrial enhanced ATP production. On the contrary, mitochondrial fission is associated with increased mitochondrial ROS generation and the uncoupled respiration for BAT thermogenesis [62,68,69]. In obese individuals, it has been shown that the fusion-fission dynamics is disrupted, leading to reduced flexibility to switch between ATP production and mitochondrial uncoupling [36]. which is essential to balance changes in nutrient availability and metabolic demands. 

Another important process in the regulation of mitochondrial dynamics and therefore, thermogenesis, is mitophagy, which selectively clears the excess of mitochondria through autophagy after damage. The principal proteins involved in this process are phosphatidylinositol-3,4,5-trisphosphate 3-phosphatase induced kinase 1 (PINK1) and p62. Mitophagy is induced by metabolic stress, including endoplasmic reticulum stress, oxidative stress, inflammation, and insulin resistance. It has been shown that the AT of obese and diabetic individuals display a disturbed mitophagy capacity [62,67], which has been associated to fat accumulation. On the other hand, in excess of mitophagy, the beige-to-white adipocyte transition is upregulated, which reduces the brown mass and therefore, the BAT characteristics upon the white ones [70,71]. The ambivalence of mitophagy in AT is necessary for the maintenance of the mitochondrial population and the tissue function [67].

### 5.2. Adrenergic Nervous System Activation of Thermogenesis Upon Cold Exposure

The brain controls many different energetic signals, including thermogenesis through the sympathetic nervous system (SNS). Its activation causes the release of norepinephrine (NE) which binds to β-3-adrenergic receptors (β3-AR) in adipocytes to activate thermogenesis [61]. Cold exposure causes the activation of specific channels located in thermoreceptor neurons innervating the surface of the body, thereby signaling for the release of NE (Figure 5). NE binds β3-AR activating adenylate cyclase, which also activates cAMP-dependent protein kinase A which promotes lipolysis and mitochondrial biogenesis through the expression of several genes including lipases, AMPK, UCP1 and PGC1α (Figure 5) [48,72].

After cold exposition, the β3-AR stimulation augments the release of fatty acids and increases the expression of CPT1a, and UCP1 for further heat production. However, the classical BAT has a higher basal UCP1 expression levels and elevated uncoupled respiration compared to beige or WAT counterparts. Although, upon hormonal stimulation with β3-AR agonists, beige cells can also elevate their UCP1 levels to the ones seen in BAT. This results suggest that beige cells are programmed to switch their function between energy storage or heat production depending on different signals [57]. In this regard, it is well known that circulating hormones, such as triiodothyronine (T3), are implicated in BAT activation and WAT browning. T3 induces the local formation of BMP8b, a BAT factor that makes this tissue more sensitive for adrenergic signaling, and also enhances SNS signaling from the brain to the BAT [72,73].

In addition, cold exposure increases lipoprotein lipase and membrane proteins implicated in the lipid uptake of plasma triglycerides or free fatty acids in BAT. Cold exposure also increases the up-regulation and translocation of glucose transporters, glucose transporter (GLUT) 1 and 4, facilitating the uptake of glucose from plasma (Figure 5). All in all, this axis provides protection against hypertriglyceridemia, hyperglycemia and insulin resistance, which are common disorders associated to obesity [10]. Additionally, cold exposure produces an increase in the production of vascular endothelial growth factor (VEGF) (Figure 5) which enhances angiogenesis and provides a venue for heat dissipation. Moreover, it has been reported that the increase in VEGF causes an increase in UCP1 expression in WAT and the expansion and activation of BAT [74].

FGF21 is also induced by cold and adrenergic activation through a cAMP-dependent mechanism (Figure 5). It is expressed mostly in the liver, adipocytes and skeletal muscle, correlating positively with the increase in EE and thermogenesis activation, and therefore, promoting browning within the AT [56,75].

Moreover, another SNS regulated gene upon cold exposure is leptin (*LEP)*, a hormone predominantly secreted from subcutaneous AT. This adipokine has multiple functions especially implicated in the control of appetite, fat storage, and glucose homeostasis. Nonetheless, although cold reduces reversibly its expression and increases browning and thermogenesis (Figure 5), it appears to exist a balance between leptin production and thermogenesis. Leptin has been shown to increase lipolysis and browning too, although much less than upon SNS activation, and many of the hypothalamic neurons involved in the regulation of thermogenesis are also leptin sensitive [76,77]. Therefore, taking all together, it appears that leptin has a double role in promoting browning and lipolysis, but after cold exposure the decreased in leptin expression is compensated by other thermogenic drivers indicated before.

## 6. Skeletal Muscle Potential in Energy Expenditure and Heat Production

### 6.1. Skeletal Muscle Features and Functions

Muscles are important metabolic organs and the main source of amino acids, storing up to 75% of the total proteins within the body. Amino acids are released when needed elsewhere in the organism. Besides, the muscle has the ability of storing glucose in the form of glycogen, being the principal source of energy for the rapid initiation of contraction even when glucose is not readily available from the diet [78].

Moreover, muscles are composed of two main types of fibers, with marked differences in their metabolic profile in regard to the speed and manner in which they metabolize glucose. The names ”fast” and ”slow” fibers, indicate the type of glucose metabolism occurring within them. The fast ones use the anaerobic metabolism of glycolysis for the quick ATP generation upon rapid contraction, and therefore they fatigue sooner. On the contrary, slow fibers are aerobic and high oxidative allowing them to be high resistant to fatigue. This second type of fibers preferentially use fatty acids as substrates for ATP production [79]. However, a remarkable feature within the skeletal muscle is the ability to a fast-shift in substrate usage between fat and glucose, depending on the needs of other organs. Therefore, the muscle is turned into a sensor of the global energy state of the body, with the ability to balance the use of different nutrients for energy dissipation or for their storage for the further use by other tissues [80].

However, when the lipid storage capacity in AT is overcome, free fatty acids are released and accumulated by other organs such as muscle and liver. Lipotoxicity in muscle impairs the contractility capacity and function of skeletal muscle, a condition named sarcopenia. This condition is increased in obese subjects as consequence of the excess of lipids excess, being worst in fast fibers and being aggravated in aging [81]. Thus, bearing in mind the important crosstalk between these the AT and muscle, the activation of the thermogenic signaling between these two organs may contribute to control the excess of lipids.

### 6.2. Thermogenesis within Skeletal Muscle

Skeletal muscle metabolism can also be activated by cold. As shown in Figure 2, BAT and skeletal muscle are derived from a common progenitor (Myf5+). Besides, it is well known that skeletal muscle is a metabolic organ which produces heat upon contraction in the process of shivering thermogenesis, thus, it is not surprising that AT and muscle might also share the capacity to accomplish the non-shivering one [60], as proton leak is also present in skeletal muscle through another uncoupling protein, the UCP3. However, while UCP3 can uncouple oxidative phosphorylation and dissipate energy as heat, this effect is secondary to its primary role which is the control of mitochondrial ROS and fatty acid oxidation (Figure 4) [10,82]. UCP3 lowers mitochondrial membrane potential and protects muscle cells against an overload of fatty acids, reducing the stress caused by excessive ROS production. In this context, SIRT1 acts as a major repressor of the UCP3 gene expression in response to glucocorticoids, which are activators of SIRT1. This requires its deacetylase activity and results in histone deacetylation in the UCP3 promoter [83], therefore controlling its levels.

Although skeletal muscle is prepared for mitochondrial non-shivering thermogenesis, UCP3 is not totally in charge of it. An alternative mechanism involving Ca^2+^ changes within the sarcoplasmic reticulum exists, which does not interfere with the mitochondrial ATP synthesis necessary for slow and fast contraction. This mechanism mainly occurs in oxidative slow fibers, through the ion pump sarcoplasmic reticulum calcium ATPase (SERCA). In this mechanism, sarcolipin, a small peptide which senses calcium levels, has been shown to activate SERCA after an influx of Ca^2+^ (Figure 4). Hence, sarcolipin promotes SERCA dependent uncoupling causing a futile cycling in the flux of Ca^2+^ through the reticulum, and promoting the dissipation of energy in the form of heat [5,84,85]. Here, with the rise of cytosolic Ca^2+^, sarcolipin promotes Ca^+2^ to concentrate at the cytoplasmic side of the membrane, instead of at the luminal side, and hence, promoting the hydrolysis of ATP producing heat. In addition, the adenosine diphosphate (ADP) produced is phosphorylated again by the oxidative phosphorylation at mitochondria which also contributes to the non-shivering thermogenesis in the muscle.

This mechanism also implicates the increased expression of PPARγ and PGC1α in muscle, which can also be induced by cold exposure (Figure 5), increasing the mitochondrial biogenesis necessary for the increased thermogenic capacity within the fibers [4,86]. Taking all of this into account, heat will be generated by SERCA during both shivering and non-shivering thermogenesis, when Ca^2+^ pumping is coupled to myofibril contraction and when it is uncoupled.

### 6.3. Exercise Performance as a Molecular Inductor of Thermogenesis and Browning

Skeletal muscle -an organ that provides us with physical force- also changes upon exercise training which also provides multiple benefits for human health maintenance and improvement. Therefore, it is not surprising that exercise endurance capacity interferes in the thermogenesic and oxidative capacity within the fibers. Exercise increases sarcolipin to produce a shift in the oxidative capacity of the fibers and hence, enhances thermogenesis. Moreover, exercise can also change the mitochondrial fate upon aging or disease by augmenting its function, oxidative capacity, mitochondrial DNA content and biomass, as well as increasing the expression of several mitochondrial biogenesis related transcription factors such as TFAM and PGC1α [87,88]. Additionally, AMPK and SIRT1 are also activated upon exercise, increasing the systemic insulin sensitivity by their role in the increase of mitochondrial biogenesis, oxidative phosphorylation and density of fibers in the skeletal muscle (Figure 4) [49,89]. Thus, the increase of the non-shivering thermogenesis in skeletal muscle is another strategy for preventing metabolic diseases.

Besides that, exercise has been shown to induce markers of BAT in WAT, augmenting the number of beige adipocytes. Skeletal muscle, which can also be considered as an endocrine organ, stablishes a communication network with other organs, such as AT. This crosstalk is mediated by several myokines, released from muscles, such as irisin, implicated in the initiation of the thermogenic and browning program in AT [10,67,90]. Irisin is proteolytically cleaved from fibronectin Type III Domain Containing 5 (FNDC5) by a PGC1α dependent mechanism and then it is released into the blood system producing a browning effect in WAT [32,58]. Another important myokine is BDNF, a factor produced by the nervous system (Figure 5) and by muscles, (Figure 4) which is able to induce lipolysis in AT, and neurogenesis in the brain. BDNF expression within the skeletal muscle is enhanced by irisin and by interleukin 6 (IL6) upon exercise (Figure 4), being the latter an important cytokine which also may act as a myokine to induce browning and lipolysis in WAT ([32,91,92].

Hence, increasing exercise, the release of browning myokines, and the skeletal muscle non-shivering thermogenesis, may be contribute to restore the metabolic homeostasis within obesity or other metabolic disorders [93,94].

## 7. Results, Discussion and Conclusions 

### 7.1. Phytochemicals as Thermogenic and Anti-Adipogenic Agents

As shown before, non-shivering thermogenesis is a potential complementary approach to prevent and/or treat obesity-related disorders [14,95]. Indeed, in the last years, there have been reported several plant-derived bioactive compounds with the capacity to activate and to augment thermogenesis. Therefore, the upregulation of several signaling pathways including UCP1, AMPK, PGC1α, Sirt1, PRDM16 and/or PPARs have been shown to increase the WAT browning [96,97,98,99,100,101,102,103,104,105].

Here, we review the evidence upon this field of some of some of the most studied natural extracts in the activation of thermogenesis.

#### 7.1.1. Pomegranate

Most health benefits of pomegranate are associated to the presence of ellagitannins, mainly punicalagin and ellagic acid [106]. Preclinical models and intervention studies with humans have shown that pomegranate, or its bioactive compound punicalagin, are able to reduce the negative effects upon high fat diet (HFD) in mice, reducing cholesterol, triglycerides and glucose in plasma, as well as the weight gain in obese individuals [107,108]. Moreover, punicalagin has the capacity of promoting mitochondrial function in vitro and in vivo, by mean of the activation of the AMPK pathway upregulating the mitochondrial biogenesis and ameliorating the oxidative stress and inflammation after a HFD [109,110]. Pomegranate extract and juice are also effective on increasing the vascular endothelial nitric oxide synthase and plasma nitric oxide levels, which in turn, increase the response to acetylcholine in vitro, which has been proposed to be beneficial against metabolic syndrome [106].

Moreover, pomegranate also contents potent antioxidants, capable of reducing lipid peroxidation, and acting as hypotensive agents to reduce blood pressure [111]. Moreover, there are several intervention studies in humans that report benefits related to the improvement of insulin sensitive or reducing weight and fat mass gain [112,113].

Furthermore, it is known that obesity is associated with immune dysfunction and a state of chronic inflammation, features that are reduced by exercise. In this regard, it has been reported that pomegranate extract in combination with exercise improves immune function in HFD-fed rats, restoring immunomodulatory factors in serum, inhibiting inflammation and decreasing oxidative stress, compared to the extract or the exercise alone [114]. Besides, it has been demonstrated that Pomegranate juice slightly reduces muscle damage markers, fatigue, and the recuperation time in elite athletes, which provides additional benefits against skeletal muscle damage and cachexia [115,116].

Punicalagin and urolithin A exert anti-adipogenic properties associated to a reduction in triglyceride accumulation and to the expression levels of adiponectin, PPARγ, GLUT4, and FABP4 in early steps of 3T3-L1 adipocytes differentiation [117]. Moreover, urolithin A increases the EE in mice by enhancing thermogenesis and browning in BAT and WAT, by mean of the elevation of triiodothyronine levels [118]. 

#### 7.1.2. Ginkgo Biloba

Ginkgo Biloba extract, a mixture of polyphenols with antioxidant properties, might be efficient in the prevention and treatment of obesity associated disorders such as insulin resistance and adipocyte hypertrophy as indicated by a variety of experimental models of endocrine dysfunctions [119]. In this regard, it have been reported several in vivo preclinical studies where Ginkgo Biloba extract significantly reduced food intake, body weight gain and adiposity, reducing the epididymal adipocyte volume, while protecting against hyperglycemia and dyslipidemia in diet-induced obesity in rats [120,121,122]. Moreover, this extract increased AMPK and adiponectin signaling pathways [123].

The anti-adipogenic effects of extracts derived from Ginkgo Biloba have been demonstrated [124]. For instance, bilobalide, a sesquiterpene compound from Ginkgo Biloba leaves [125], or Ginkgetin, a Biflavone from Ginkgo Biloba leaves [126], blocked the differentiation of preadipocytes into adipocytes, reducing PPARγ and C/EBPα expression in 3T3-L1 adipocytes during adipogenesis. In addition, they also increased lipolysis by activating AMPK signaling pathway and the expression of CPT1α, which contributed to reduce hypertrophy of WAT in HFD mice. Thus, suggesting a potentially anti-obesogenic effect in longer-term therapies.

Another bioactive extract derived from the seed coat of Ginkgo Biloba have also been investigated. Although this part is rarely use and it is typically discarded, ginkgo vinegar from the seed coat has been reported to suppress the expression of C/EBPδ and PPARγ, key proteins in adipogenesis, and to inhibit lipid accumulation in 3T3-L1 cells that were induced to become adipocytes, hence, inhibiting adipocyte differentiation [127].

#### 7.1.3. Milk Thistle

Silymarin, which belongs to the flavonolignan group, is the main bioactive constituent found in Milk Thistle (*Silybum marianum*). Silymarin has been shown to exhibit antioxidant, plasma lipid-lowering effects, antihypertensive, antidiabetic, antiatherosclerotic, anti-obesity, and hepatoprotective effects [128,129].

The beneficial effects have been related to the increase of genes implicated in the promotion of BAT (Sirt-1, PPARα, PGC-1α, and UCPs) and to the decrease on the expression of genes related to WAT differentiation (PPARγ, FABP4, FASN, SREBP1c, C/EBPα) [130,131]. In addition, silymarin has been shown to diminish lipid accumulation and early adipogenesis via the regulation of cell cycle and AMPK signaling pathways in vitro [132]. Therefore, by mean of the induction of thermogenesis and the promotion of brown remodeling in adipocytes, silymarin reduced fatty acid accumulation and adipocyte size. Besides that, in a recent preclinical assay in obese mice, Silymarin reversed the AT inflammation and adipocyte hypertrophy, stopping weight gain without changes in food intake. In addition, it reversed liver disorders restoring insulin sensitivity, and glucose and lipid homeostasis [133].

In conclusion, silymarin and silybin, its major active constituent, have important roles in the treatment and prevention of obesity through several mechanisms including the suppression of the expression of adipogenesis-related genes, and the reduction of lipid mass but augmenting the functional capacity of adipocytes [129].

#### 7.1.4. Soy

Because of the lower frequency of obesity and diabetes II diseases in Asian countries, attention has been turned toward their diet, which consists in high amounts of soy and soy-based products. Their principal bioactive components are isoflavones, such as genistein and daidzein, which are similar in structure to endogenous estrogens [134]. They interact with estrogen receptors, which results in the reduction of intracellular lipids. Therefore, although various studies have focused on the phytoestrogenic function of isoflavones, their potential to increase WAT browning and non-shivering thermogenesis has been investigated [135,136].

Besides protein may also be particularly effective in preventing diet-induced-obesity. protein has been reported to prevent fat mass gain under HFD, inducing browning in WAT and lipolysis and thermogenesis in BAT, by mean of the increase of UCP1 expression and the leptin sensitivity in the hypothalamus [137,138] found that diets rich in Isoflavones also increased triiodothyronine levels and UCP1 mRNA levels in the BAT of rats, although the core body temperature decreased. Isoflavones have demonstrated to enhance mitochondrial biogenesis in AT through the SIRT1/PGC1α pathway [137]. 

Soy ssoflavones are known to exert lipolytic and anti-adipogenic effects in WAT, alone or in combination with other bioactive compounds. For instance, they have an additive, not synergic, anti-adipogenic effect in combination with green tea extract and grape resveratrol, reducing the expression of PPARγ, C/EBPα and FABP4 among others, in in vitro studies with 3T3-L1 and human adipocytes [138]. Daidzein alone also suppressed adipogenesis in 3T3-L1 preadipocytes [139]. Another study also showed that daidzein and genistein inhibited adipogenesis in human adipocytes reducing the expression of WAT markers, PPARγ, SREBP-1, FASN, C/EBPα, although by mean of different mechanisms of action, as daidzein inhibited adipogenesis through the stimulation of lipolysis, and genistein inhibited the glycerol-3-phosphate dehydrogenase activity acting via the activation of estrogen receptors [140,141]. In contrast to this results, [142] showed in vitro that Genistein and Daidzein functioned as PPARγ agonists.

Genistein has been shown to induce BAT adipogenesis in 3T3-L1 preadipocytes when treated at initial points of the differentiation process. In another study, genistein induced the expression of BAT specific markers (UCP1, SIRT1 and PGC1α), reduced WAT markers (FASN, FABP4), and increased the mitochondrial proton leak and oxygen consumption, promoting features of beige adipocytes [97]. Genistein also increased the body temperature and plasma levels of triiodothyronine of obese mice [143]. 

In addition, genistein exerted a dose-dependent effect on adipocyte differentiation and function being able to control adiposity [144]. Genistein and daidzein increased PGC-1β gene expression and augmented the EE energy in a preclinical model of HFD induced obesity [145]. Besides that, daidzein reduced weight gain and fat content in liver, which was associated to the increase of fatty acids oxidation and to the enhanced expression of UCP1 in BAT [146,147]. 

Usually, obesity also involves the ectopic accumulation of lipids in the skeletal muscle. Genistein enhanced fatty acid oxidation in muscle, by mean of AMPK dependent mechanism, and increased the expression of PGC1α and PPARδ, through a mechanism that involved cAMP in a leptin receptor-independent manner [148]. Daidzein enhanced TFAM expression through the SIRT1/PGC1a pathway to promote mitochondrial biogenesis in muscle [149]. In addition, Soy Isoflavones have been shown to protect against muscle atrophy through a SIRT1 dependent mechanism [150], augmenting the diameter and number of mitotubes, and increasing the expression of insulin growth factor and myosin heavy chain in C2C12 muscle cells [151]. They also increased soleus muscle mass, but only in female mice, a feature that could be related to its similarity in structure with estrogens [152].

#### 7.1.5. Resveratrol

Resveratrol is a non-flavonoid polyphenol present in grapes and other food vegetables. It is an antioxidant and has anti-inflammatory effects that can improve mitochondrial biogenesis and fat browning. Resveratrol induced thermogenesis in BAT, reduced fat accumulation in WAT, and increased lipolysis in liver and muscle, and improved aerobic respiratory capacity in muscle cells in vitro. [153,154]. It is one of the most studied bioactive compounds regarding plant-based food products, thus, many studies have reported its effects in several fields such as aging, obesity, immunology and cancer [155]. However, there is still a lack of satisfactory results in human studies regarding its effects in the activation of thermogenesis and body weight management [156,157].

In this context, resveratrol supplementation increased the oxidative capacity and mtDNA content, and reduced lipogenesis and insulin resistance markers in vitro [158,159]. Many studies have shown in vivo that resveratrol supplementation reduced body fat, as well as increased thermogenesis by activating BAT activity and mitochondrial function. Resveratrol is a natural activator of sirtuin, and it has been demonstrated to promote the expression of lipolytic and thermogenic genes like UCP1 and PRDM16, to augment mitochondrial biogenesis by mean of the increase expression of PGC1α, SIRT1, TFAM in BAT [160,161], and to increase UCP3 and TFAM expression in muscle of obese animals [162]. In non-obese mice, resveratrol improved insulin resistance and mitochondrial function in muscles, suggesting benefits in the prevention of metabolic disorders [163,164]. Resveratrol also stimulated mitochondrial fusion (increasing mitofusin-2 expression, which enhanced the mitochondrial mass within cells, and thus, improved the oxidative respiration and thermogenic capacity [165]. Moreover, in a model of metabolic syndrome in rats, the supplementation of Resveratrol in combination with quercetin improved UCP2 expression in WAT, raising the levels of oleic and linoleic fatty acids to activate PPARα [166]. 

In the same line, pterostilbene, a dimethyl ether derivative from Resveratrol, had similar effects regarding BAT thermogenesis and WAT browning by mean of SIRT1 dependent activation of AMPK to increase PGC1α, CPT1α and UCP1 expression levels [167,168]. In addition, AMPK activation induced WAT browning in in vivo models of obesity [159,169]. Resveratrol promoted lipolysis in SGBS human and 3T3L1 murine adipocytes in vitro, as well as in white AT from mice, by mean of the increased expression of adipose triglyceride lipase (ATGL) levels [170] which reduced fat content and body weight after an obesogenic diet. 

Furthermore, [171] showed that the supplementation of mice mothers during pregnancy and lactation increased browning and thermogenesis in WAT of mice after weaning. Nevertheless, recently it has been reported that this browning in WAT tissues upon the supplementation with resveratrol is sex-dependent, as it is significative more notorious in the males´ primary neonatal adipocytes than in the females’ ones [172]. In line with this, supplementation with resveratrol in early steps of life positively affected the browning process of adulthood WAT in males when exposed to HFD [173].

As commented above, another important protein in thermogenesis is irisin, which increases the expression of browning markers. In this context, it has also been reported in primary subcutaneous adipocytes from humans that FNDC5, the peptide that releases irisin after proteolysis, is highly increased after resveratrol supplementation [174]. However, the in vitro treatment of C2C12 derived myotubes with resveratrol did not increased FNDC5 expression [175]. 

Resveratrol decreased adipogenesis in 3T3-L1 preadipocytes in a dose-dependent manner [176,177], and in human visceral derived preadipocytes [99]. These effects were mediated through AMPK-SIRT1 pathway which decreased the expression of PPARγ, C/EBPα, SREBP1c and fatty acid synthase (FAS), and inhibited insulin signaling, mitochondrial biogenesis, and lipogenesis in preadipocytes [178,179]. In vivo resveratrol inhibited the visceral adipogenesis and inflammation in HFD-fed mice [180]. Resveratrol also decreased lipogenesic markers in mice treated with high protein diets [181]. Additionally, resveratrol improved the plasmatic levels of lipids and glucose in mice treated with standard diets [182].

Besides its actions in AT, Resveratrol has benefits in muscle tissues promoting increased oxidative capacity. For instance, in a human study, resveratrol supplementation increased the running time and consumption of oxygen in muscle fibers, augmenting markers of oxidative phosphorylation and mitochondrial biogenesis, such as PGC1α activity and SIRT1, as well as key regulators of energy and metabolic homeostasis [183]. Resveratrol improved the mitochondrial function and biogenesis, through the SIRT1/PGC-1 pathway, in an in vitro model of endothelial cells and in the aortas of type 2 diabetic mice [184]. Additionally, in humans, a low dose of resveratrol supplementation activated the SIRT1/PGC1α pathway in skeletal muscle, improving mitochondrial function, although no changes in BAT were observed [185]. This could be attributed to interindividual differences in the metabolism of resveratrol, and/or due to differences in the BAT content between humans and animals. 

Moreover, resveratrol also increased muscle aerobic capacity, and reduced fatigue [186]. It is known that exercise training improves the endurance capacity of muscles by increasing both mitochondrial number and function. Resveratrol supplementation has been suggested to enhance the physical performance due to its effect to augment the capacity for fatty acid oxidation [187]. Resveratrol supplementation in humans has been shown to improve muscle glycogen content, insulin sensitivity, and to reduce muscle hypertrophy and muscle fatigue in combination with exercise [188], and in elderly humans [189], suggesting an anabolic role in exercise-induced adaptations. Although [190] did not find the same effect in this regard in aging mice. 

Based on these findings, the use of polyphenols like resveratrol to enhance UCP1-dependent and independent thermogenesis in BAT, and the enhanced capacity to augment the oxidative capacity of muscles are promising strategies to alleviate the metabolic stress associated to chronic diseases such as obesity. 

Table 1 summarizes main studies describing the antiobesity and thermogenic effects of pomegranate, silymarin, ginkgo and resveratrol extracts and their molecular mechanism of action.

### 7.2. Relevance of Research on Bioactive Compounds to Augment Energy Expenditure

Obesity and its associated metabolic disorders are currently a serious problem in the world population. Numerous treatment alternatives to combat different aspects of metabolic alterations related to obesity are currently being studied. Thus, in this review, we propose to increase EE by promoting a correct balance of nutrients in a personalized way, throughout the induction of adaptive thermogenesis in adipose tissue and muscle by phytochemicals. Bioactive compounds from natural sources or diet derived ingredients can ameliorate metabolic and oxidative stress by targeting relevant pathways including activation of AMPK/PGC1a/SIRT1, involved in mitochondrial biogenesis and activation of thermogenesis (which increases aerobic capacity in muscle and browning in adipose tissue); the induction mitochondrial uncoupling through the upregulation of UCP1, the reduction of pro-inflammatory markers such as IL6 and TNFa, among others features. Therefore, bioactive compounds can provide complementary approaches in the treatment of obesity and associated metabolic alterations including insulin resistance, hyperglycemia, hypercholesterolemia, hypertension, dyslipidemias, low grade of chronic inflammation and even cancer. However, few studies have evaluated the potential of bioactive compounds to promote energy expenditure in clinical trials in humans. Thus, this revision may provide relevant information to develop diets and supplements to conduct nutritional intervention studies with the objective to ameliorate the metabolic and chronic inflammation in the course of obesity and related disorders. 

### 7.3. Concluding Remarks

In conclusion, as many processes implicated in obesity, type 2 diabetes, insulin resistance, and metabolic syndrome are related to AT dysfunction and skeletal muscle atrophy, it is highly important to know the molecular pathways implicated in their processes. Thus, with this knowledge, a nutrigenomic personalized strategy could be accomplished if those pathways are clear, and if we know how different nutrients could affect them. Therefore, as natural bioactive compounds are considered an excellent alternative strategy for developing safe and cost-effective anti-obesity agents, we should make rapid and substantial progress in the discovery of new inducers of thermogenic natural phytochemicals for precision nutrition strategies in the upcoming years. Thus, future studies are needed for clarifying important processes related with those disorders in a genetic nutrition context.

## Figures and Tables

**Figure 1 cancers-13-00866-f001:**
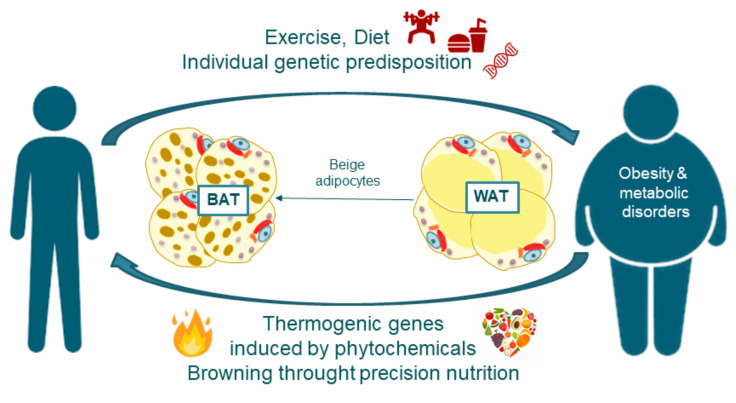
Fat burning non-shivering thermogenesis and browning through precision nutrition as mechanisms to prevent and treat obesity associated disorders.

**Figure 2 cancers-13-00866-f002:**
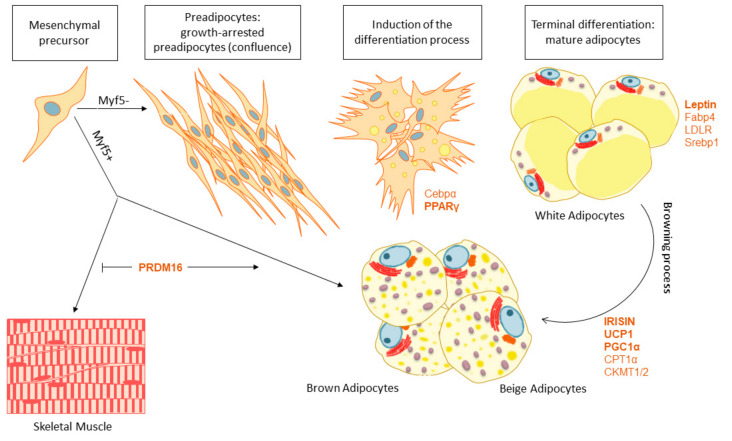
Adipogenesis, the differentiation process from the mesenchymal precursors to BAT or WAT and the browning process, main genes implicated.

**Figure 3 cancers-13-00866-f003:**
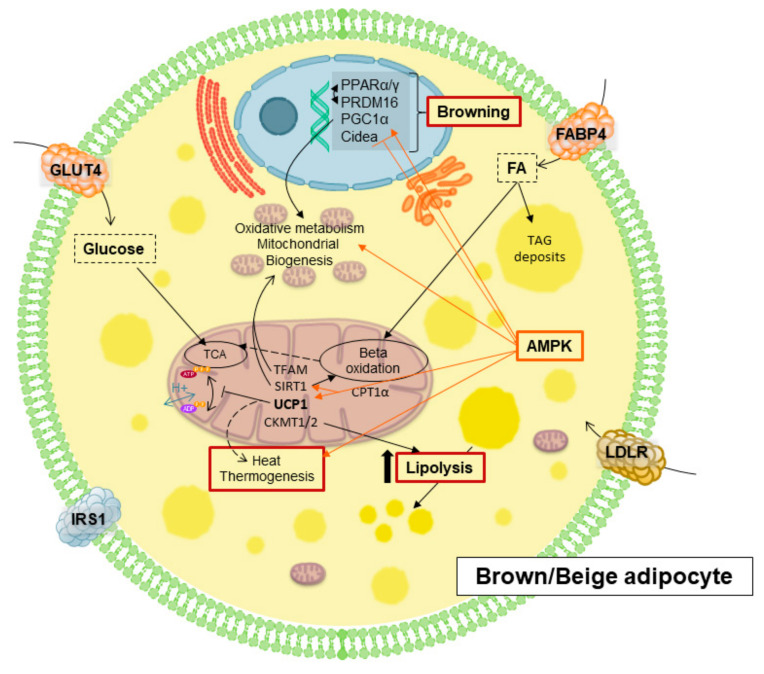
Main mediators implicated of browning and thermogenesis in brown/beige adipocytes.

**Figure 4 cancers-13-00866-f004:**
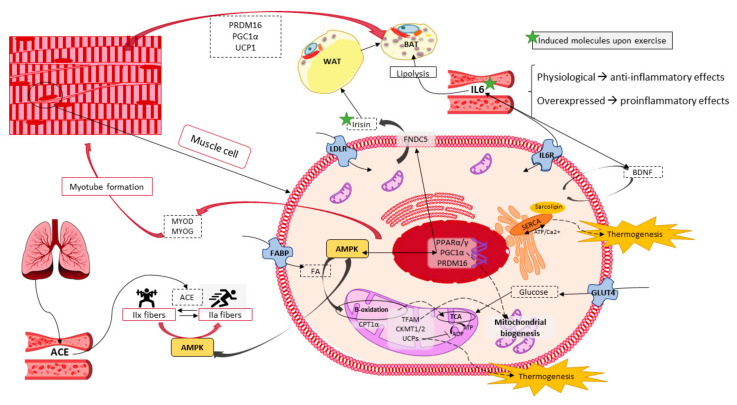
Skeletal muscle potential in energy balance through thermogenesis, and its influence on the browning and fiber-switch processes. Genes and proteins implicated.

**Figure 5 cancers-13-00866-f005:**
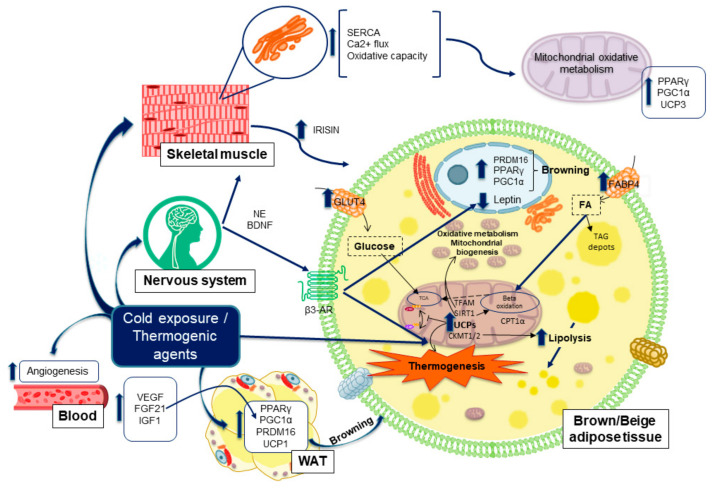
Activation of BAT thermogenesis and WAT browning upon cold exposure: role of the SNS.

**Table 1 cancers-13-00866-t001:** Antiobesity and thermogenic effects of pomegranate, silymarin, ginkgo and resveratrol extracts and their potential molecular mechanism of action.

Ref.	Extract	Treatment	Protocol	Effects
[117]	Pomegranate extract	Urolithin A	in vitro (3T3L1)	↑ adipogenesis, TG, lipase, PPARG, GLUT4, FABP4, adiponectin
[107]		Punicalagin	in vivo (HFD mice 400–800 mg/kg/d 5 wk)	↓ Ch, TG, glc, BW, appetite
[110]		Punicalagin	in vivo (HFD mice 150 mg/kg/d)	↑ AMPK, PGC1a pathway
[109]		Punicalagin	in vivo (HFD mice 150 mg/kg/d)	↓ oxidative stress, inflammation markers (IL1, 4, 6, TNFa), hyperlipidemia, hepatic lipid deposition ↑ FAO, PGC1a
[113]		Seed oil	in vivo (HFD mice 1% diet, 12 wk)	↑ Insulin sensitivity ↓ BW, WAT mass
[114]		Pomegranate extract + exercise training	in vivo (HDF rat 150 mg/kg/d 8 wk, training 60’ 3 times/wk)	↑ immune function (CD4+) ↓ apoptosis in PBMC, inflammation, oxidative stress
[118]		Urolithin A	in vivo (mice 30 mg/kg/d 10 wk)	↑ Browning and thermogenesis (T3), improves glucose and insulin homeostasis. ↓ BW
[115]		Pomegranate juice	Prospective cross open cohort-controlled study (athletes 1.5 L/d 2 days with power training)	↓ muscle damage markers, fatigue, recuperation time
[123]	Ginkgo Biloba	Isoginkgetin	in vitro (3T3L1)	↑ AMPK, adiponectin pathways, no effects in PPARG nor adipogenesis
[125]		Bilobalide	in vitro (3T3L1)	Antiadipogenic effects ↓ differentiation, TG accumulation, ↑ AMPK, CPT1a, HSL, lipolysis
[127]		Vinegar from seed coat	in vitro (3T3L1)	↓ Lipid accumulation, adipogenesis and differentiation (Cebpd, PPARG)
[191]		Bilobalide	in vitro (3T3L1)	↓ NFkb ↑ Adiponectin
[124]		Ginkgo Biloba extract	in vitro (primary mice adipocytes and osteoblasts) and in vivo (hamster HCD/HFD 30 d 250 mg/kg/d)	Antiadipogenic effects ↓ PPARG, Ch. ↑ Apoptosis via ROS in WAT
[126]		Ginkgetin	in vitro (3T3L1) and in vivo (mice HFD 5–10 mg/kg/d)	Antiadipogenic. ↓ differentiation, STAT5, PPARG, Cebpa ↑ hypertrophy AT in mice
[121]		Ginkgo Biloba extract	in vivo (rat 2 months HFD + 14 days 500 mg/kg/d)	↓ Intake, BW, NFkb, TNFa IR ↑ IL10, Akt-P
[120]		Ginkgo Biloba extract	in vivo (rat 2 months HFD + 14 days 500 mg/kg/d)	↓ Intake, IR ↑ Akt-P, IRS1
[192]		Ginkgo Biloba extract	in vivo (hypertensive rats 3 wk 100 mg/kg/d)	↓ BP, Nitrite level ↑ eNOS mRNA, iNOS prot, TNFa, IL6, IL1, GSH
[131]	Milk Thistle Silymarin	Silibinin	in vitro (3T3L1)	↓ PPARG, FABP4, FASN, SREBP1c, Cebpa en WAT, terminal differentiation, lipogenesis in mature adipocytes
[130]			in vitro (mesenquimal stem human adipocytes)	↓ PPARG, FABP4, FASN, SREBP1c, Cebpa en WAT ↑ SIRT1, PGC1a, UCP1
[132]			in vitro (3T3L1) and in zebra fish	↓ Lipid accumulation (TG, FA), adipogenesis and differentiation (Cebpd, PPARG, FABP4), adipocyte size, ↑ AMPK
[193]			in vivo (rat 49–77 d HFD 200 mg/kg/d)	↓ BMI, IR, TG, LDL ↑ Leptin sensitivity
[128]			in vivo (mice HFD 18 d 30–60 mg/kg/d)	↓ Lipid accumulation, IR, BP, BW, inflammation Improve glucose metabolism
[194]			in vivo (rat 4 wk 1% Silymarin in diet)	↑ HDL, ABC transporter, CytP450 ↓ TG, Ch in serum
[133]			in vivo (obese mice, 8 wk HFD + 8 wk 50 mg/kg/d intraperitoneal)	↓ AT inflammation, hypertrophia, BW, IR, restore lipid and glucose homeostasis
[195]			in vivo (rar 42 d 26 mg/kg/d)	↑ Serum lipid profile, SOD, GSH, Adiponectin, FAO ↓ IR, Resistin, Oxidative stress, FA synthesis
[196]			in vivo (rat HFD, 6 wk 0.5 mg/kg/d)	↓ IR, visceral fat, gluconeogenesis, TG ↑ Lipolysis
[197]	Soy	Mix of Soy Isoflavones	in vitro	Antiadipogenesis, ↓ SREBP1c
[198]		Genistein	in vitro (primary human adipocytes)	↓ Cebpa, PPARG, LPL, Lipid droplet size ↑ TGFb1
[141]		Genistein	in vitro (primary human adipocytes)	↓ Adipogenesis and differentiation (PPARG, Cebpa, FABP4, FASN, SREBP1c)
[140]		Genistein and Daidzein	in vitro (human derived mesenquimal stem cells)	↓ adipocyte differentiation (PPARG, Cebpa, SREBP1c, GLUT4)
[148]		Genistein	in vitro (3T3L1) and in vivo (mice 0.2% Genistein in diet 58 d)	↑ FAO, browning induction (FNDC5) mitochondrial function in mice muscle (AMPK, PGC1a, PPARG) ↑ thermogenesis (UCP1, TMEM16), mitochondrial number and respiration rate in adipocytes 3T3L1
[199]		Genistein	in vitro (primary epididimal rat adipocytes)	↑ Lipolysis, cAMP via AMPK activation ↓ TG
[200]		Genistein	in vitro (3T3L1)	↑ AMPK, apoptosis in mature adipocytes ↓ adipogenesis
[137]		Mix of Soy Isoflavones	in vitro (primary adipocytes)	↑ mitochondrial biogenesis (SIRT1-PGC1a pathway), ATP synthase b
[138]		Soy Isoflavones + Green Tea + Resveratrol	in vitro (3T3L1)	↓ adipogenesis and differentation (PPARG, Cebpa, FABP4 and perilipin)
[139]		Daidzein	in vitro (3T3L1)	↓ Adipogenesis (PPARG, Cebpa), lipid accumulation, PI3K-Akt pathway
[97]		Genistein	in vitro (3T3L1)	↑ thermogenesis in BAT (UCP1, SIRT1, PGC1a, proton leak and oxygen consumption) ↓ Lipid accumulation in WAT (FASN, FABP4, HSL, resistin)
[149]		Daidzein	in vitro (C2C12)	↑ mitochondrial biogenesis (PGC1a, TFAM, SIRT1 dependent), COX1
[150]		Mix of Soy Isoflavones	In vitro (C2C12)	↑ SIRT1, AMPK activation ↓ myotube atrophy
[151]		Mix of Soy Isoflavones	in vitro (C2C12)	↑ myotube diameter, MHC protein, IGF1 and IGF1R
[201]		Genistein	in vivo (mice 0–1500 mg/kg/d 3 wk)	↑ fat tissue apoptosis ↓ food intake, BW, parametrial and inguinal fat
[136]		Mix of Soy Isoflavones	in vivo (rat HFD 8 wk HFD + 4 wk HFD + 50–400 mg/kg/d)	↓ BW, lipogenesis, adipogenesis ↑ FAO, lipolysis, Akt-P, mTOR inhibition
[202]		Genistein and Daidzein	in vivo (mice 3 wk 286 ppm geistein + 198 ppm Daidzein)	↓ BW, WAT mass, serum leptin, insulin, TG in muscle and liver ↑ AMPK, ACC, FAO, mitochondrial biogenesis (PGC1a, TFAM) in muscle and fat
[203]		Soy protein	in vivo (rat HFD 30% Soy protein 180 d)	↑ UCP1, WAT lipolysis, Leptin sensitivity in hypotalamous, adipocyte perilipin ↓ SREBP1 and adipocyte size in WAT
[204]		Mix of Soy Isoflavones	in vivo (rat 10–600 mg/kg)	↑ thermogenesis (UCP1, T3 in BAT) ↓ leptin and insulin in serum
[143]		Genistein	in vivo (obese mice 600 mg/kg/d 5 wk)	↑ body temperature, T3 in serum ↓ hypercorticosteronism
[146]		Daidzein	in vivo (obese rat 50 mg/kg/d 14 d)	↓ BW, fat in the liver, SCD ↑ FAO and UCP1 in BAT
[147]		Isoflavones and Soy protein	in vivo (rat 0–4 g/kg/d)	↑ thermogenesis and browning (UCP1,2, 3, PPARa) ↓ WAT adipogenesis (PPARG)
[205]		Isoflavones and Soy protein	Randomized placebo controlled trial (postmenopausal 160 mg/d Isoflavones + 20 g/d Soy protein 3 months)	↓ abdominal and subcutaneous fat, IL6 No effect in leptin/adiponectin
[176]	Grape Resveratrol	Resveratrol	in vitro (3T3L1)	↓ Adipogenesis (↓ adipogenesis (PPARG, Cebpa, SREBP1c, FASN) ↑ SIRT1, AMPK activation, apoptosis, TNFa and lipolysis
[165]		Resveratrol	in vitro (C2C12 myoblast, PC3 cancer cells, mouse embryonic fibroblast)	↑ mitofusin 2 expression and respiration rates
[206]		Pterostilbene	in vitro (3T3L1)	↑ adiponectin. ↓ cell proliferation and differentiation (PPARG, Cebpa, FASN and resistin)
[207]		Pterostilbene	in vitro (3T3L1)	↑ oxygenase I ↓ Differentiation (PPARG, Cebpa, FABP4)
[208]		Pterostilbene	in vitro (3T3L1)	↓ Lipogenesis and lipogenic insulin effect
[99]		Resveratrol	in vitro (3T3L1)	↓ adipogenesis and differentiation (PPARG, Cebpa, SREBP1c, FASN, FABP4) dose dependent
[178]		Resveratrol	in vitro (3T3L1, SGBS)	↑ mitochondrial biogenesis and mass (AMPK, ATAD3) ↓ lipogenesis
[179]		Resveratrol	in vitro (bovine intramuscular adipocytes)	↑ SIRT1, AMPK, FOXO1 pathways, HSL ↓ Adipogenesis (FASN, PPARG)
[170]		Resveratrol	in vitro (3T3L1, SGBS)	↑ FA release, ATGL via AMPK activation
[209]		Resveratrol	in vitro (3T3L1) and in vivo (mice HFD 1–30 mg/kg/d 10 wk)	↓ lipid deposition in WAT and liver, BW, differentiation capacity (PPARG and perilipin)
[158]		Resveratrol	in vitro (3T3L1)	↑ mtDNA, oxydative capacity (CPT1a) and thermogenesis (UCP1) ↓ Lipogenesis and resistin
[183]		Resveratrol	in vivo (HFD 15 wk 400 mg/kg/d)	↑ EE, thermogenesis (UCP1), mtDNA, mitochondrial biogenesis (PGC1a, PPARA) and oxygen consumption in muscle fibers
[162]		Resveratrol	in vivo (rat HFD 30 mg/kg/d)	↑ SIRT1, COX2, PGC1a and UCP1 protein
[168]		Resveratrol	in vivo (mice HFD + 0.04–0.4% Resveratrol 8 month)	↑ mitochondrial biogenesis and function (PGC1a, NRF2, UCP1, ATP5a1, TFAM, SIRT1, AMPK activation, and maximal respiration rate)
[163]		Resveratrol	in vivo (mice 8 wk 4 g/kg)	↑ thermogenesis and mitochondrial function (UCP1, SIRT1, BMP7)
[169]		Resveratrol	in vivo (mice HFD 4 wk 0.1% Resveratrol)	↑ iBAT mass, thermogenesis and browning (UCP1, AMPK, PRDM16)
[160]		Resveratrol	in vivo (rat ND 30 mg/kg/d 6 wk)	↑ thermogenesis and mitochondrial function (UCP1, SIRT3, ↓ PGC1a acetylation)
[159]		Resveratrol	in vivo (mice HFD 0.1% Resveratrol)	↑ thermogenesis, browning and mitochondrial function in iWAT (UCP1, PRDM16, Cidea, PGC1a, AMPK, oxygen consumption and FAO)
[161]		Resveratrol	in vivo (mice HFD 0.5% Resveratrol)	↑ thermogenesis and mitochondrial function (UCP1, PRDM16, PPARA and adiponectin expression, SIRT1 and PGC1a activation)
[164]		Resveratrol	in vivo (mice HFD/ND + 10 mg/kg/d)	↑ mitochondrial activity and mass in BAT, extrogen receptor a
[166]		Resveratrol + quercetin	in vivo (rat 4wk high glucose in water + 10–50 mg/kg/d)	↑ PPARG, UCP2 in WAT, MUFAs and PUFAs
[210]		Pterostilbene	in vivo (rat 15–30 mg/kg/d)	↑ browning and thermogenesis (UCP1, PPARA, NRF) and oxidative capacity (CPT1a)
[171]		Resveratrol	in vivo (mice HFD 0.2% Resveratrol during pregnancy and lactation/breeding 11 wk)	↑ EE, BAT function, browning and thermogenesis after weaning (UCP1, PRDM16, Cidea, PGC1a, SIRT1, AMPK) ↓ IR, TG, WAT mass, blood glucose
[172]		Resveratrol	in vivo (postnatal mice 2–20 d 2 mg/kg/d)	↑ thermogenesis in BAT only in males (UCP1, PGC1a, TMTM26, SLC27a1, CPT1b
[174]		Resveratrol	in vivo (mice 400 mg/kg/d 8 wk) and preclinical (*n* = 20, 50 mg/d)	↑ Browning and thermogenesis (UCP1, PRDM16, PGC1a SIRT1 dependent and FNDC5 in subcutaneous AT)
[180]		Resveratrol	in vivo (mice HFD 0.4% Resveratrol 10 wk)	↓ Adipogenesis (FASN, leptin, PPARG, Cebpa, SREBP1c, FABP4), inflammation (TNFa, IL6, INFa and b), TG, BW, Ch, blood glucose
[181]		Resveratrol	in vivo (mice HFD and HPD, 4 g/kg/d 60 d)	↓ adipogenesis and lipogenesis (PPARG, Cebpa, SREBP1c, FASN), BW, Ch, AT mass, ACC ↑ HDL
[186]		Resveratrol	in vivo (mice 0–125 mg/kg/d 21 d + swimming training)	↑ muscle aerobic capacity ↓ muscle fatigue, CK, ammonia, lactate in serum
[187]		Resveratrol	in vivo (rat 4 g/kg/d 12 wk + physical training)	↑ Force isometric contraction, FAO, physical performance, mitochondrial number and function (oxydative metabolism), cardiac function (FAO)
[188]		Resveratrol	in vivo (mice 25 mg/kg/d 4 wk + climbing exercise)	↓ muscle fatigue index ↑ muscle glycogen content, insulin sensitivity, muscle hypertrophy
[211]		Resveratrol	Randomized doubleblind crossover trial (11 obese men 30 d 150 mg/d)	↑ Lipolysis ↓ adipocyte size
[185]		Resveratrol	Randomized, placebo controlled, cross-over trial. (13 relatives to T2DM patients 150 mg/kg/d 30 d)	↑ SIRT1, PGC1a pathways in skeletal muscle ex vivo No changes in BAT
[212]		Resveratrol	Part of a randomized, double-blind, parallel group trial (10 men T2DM 12 wk 2 g/d)	No changes in BMI, AT mass ↑ resting EE, SIRT1, AMPK expression in muscle
[189].		Resveratrol	Randomized blind placebo-controlled trial (30 elderly subjects, 500 mg/d 12 wk + regular exercise)	↑ mitochondrial density, knee extensor muscle peak torque ↓ muscle fatigue index

Abbreviations: 4-hydroxybutyryl-CoA dehydratase HCD, Acetyl-CoA carboxylase ACC, Adenosine triphosphate ATP, Adenosine monophosphate -activated protein kinase AMPK, Adipose Tissue AT, Adipose triglyceride lipase ATGL, Antioxidant response element 2 NRF2, ATPase family AAA domain-containing protein 3 ATAD3, Blood pressure BP, Body weight BW, Bone morphogenetic protein 7 BMP7, Brown Adipose Tissue BAT, Carnitine Palmitoyltransferase 1A CPT1a, CCAAT-enhancer-binding protein α C/EBPα, Cholesterol Ch, Cyclooxygenase 1 COX1, Creatinin kinase CK, Energy expenditure EE, Fatty Acid Binding Protein 4 FABP4, Fatty acid oxydation FAO, Fatty acid synthase FAS, Fibronectin Type III Domain Containing 5 FNDC5, Forkhead box protein O1 FOXO1, Glucose transporter GLUT, Glutathione GSH, High density lipoprotein HDL, High fat diet HFD, High fat hydrocabure diet HFHD, High protein diet HPD, Hormone-sensitive lipase hsl, Interferon a & b, INFa/b, Interleukin, Insulin-like growth factor 1 IGF1, Insulin-like growth factor 1 IGF1R, Insulin receptor substrate IRS, Insulin resistance IR, Myosin heavy chain MHC, Mitochondrial DNA mtDNA, Mitochondrial Transcription Factor A TFAM, Monounsaturated fatty acids MUFA, Necrosis tumoral factor a TNFa, Nitric oxide synthase eNOS, Normal diet ND, Nuclear factor kappa-light-chain-enhancer of activated B cells NFkB, Lipoprotein lipase LPL, Low density lipoprotein LDL, Peripheral blood mononuclear cell PBMC, Peroxisome Proliferator-Activated Receptors PPARs, Peroxisome proliferator-activated receptor gamma—coactivator 1a PGC1α, Phosphatidylinositol 3-kinase PI3K, PR domain containing 16 PRDM16, Protein kinase B Akt, Polyunsaturated fatty acids PUFA, Reactive oxygen species ROS, Signal transducer and activator of transcription 5 STAT5, Sirtuin Sirt, Sterol regulatory element-binding transcription factor 1 SREBP1c, Stearoyl-CoA desaturase-1 SCD, Superoxide dismutase SOD, Transmembrane member 16 TMEM16, Triiodothyronine T3, Triglycerides TG, Uncoupling proteins UCPs, White Adipose Tissue WAT. ↑ indicates upregulation, ↓ indicates down-regulation.

## Data Availability

Not applicable.

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
