# Peer review of "Precision Nutrition to Activate Thermogenesis as a Complementary Approach to Target Obesity and Associated-Metabolic-Disorders"

_cancers, 2021, doi:10.3390/cancers13040866_

Round 1
Reviewer 1 Report
This review is very interesting, well documented, with a lot of new information.
Some remarks:
The abstract - is a bit long. Lines 521, 523 - ginkgo biloba.
Please correct with a large G.Lines 567 - The beginning of this sentence is missing: "137 found that diets rich in isoflavones also increased triiodothyronine levels and…."
To arrange the table (it has no beginning: table head);
Ginkgo Biloba (with small b); and so on The medicinal plants, extracts or active ingredients taken in this study are all numbered with the letter A. Why?Line 687 - summarizes All in all (?)
Author Response
See attached document

Reviewer 2 Report
The manuscript was prepared very well. I congratulate the authors for the preparation of the manuscript
However, I have the following comments:
- All the figures in the manuscript: The figures are of their creation? What design program was used to make it? Indicate in the figure footer all the acronyms
- Table 1: please reconstruct the table. In the 1st column include author, year and reference. In another column include the type of trial: controlled, randomized...Even if you include the extracts, do all the effects correspond to the same active principle of the extract? The table should have its own acronyms in a legend at the end. Include the table in the results section.
- Methods: It should include a full methodology section. That includes search terms (MeSH), inclusion and exclusion criteria, strategies, search dates, methodological quality of the items included in the results table...
- Even if you include different sections in the manuscript, include a results section and a discussion section. Rewrite the manuscript. In these sections you can include them in these new sections. This would make the article more understandable.
- Include a set of practical applications
- Review all acronyms in the text. The first time they appear, they should be explained
- The references do not conform to the journal's standards. Neither the numbers of citations nor the format of the references
Author Response
See attached document

Round 2
Reviewer 2 Report
I would like to congratulate the authors for the changes made and the work done in solving the suggestions. I have no further suggestions